

# Enhancing transportation network intelligence through visual scene feature clustering analysis with 3D sensors and adaptive fuzzy control

Jing Xu

Xiamen Academy of Arts and Design, Fuzhou University, Xiamen, Fujian, China

## ABSTRACT

The complex environments and unpredictable states within transportation networks have a significant impact on their operations. To enhance the level of intelligence in transportation networks, we propose a visual scene feature clustering analysis method based on 3D sensors and adaptive fuzzy control to address the various complex environments encountered. Firstly, we construct a feature extraction framework for visual scenes using 3D sensors and employ a series of feature processing operators to repair cracks and noise in the images. Subsequently, we introduce a feature aggregation approach based on an adaptive fuzzy control algorithm to carefully screen the preprocessed features. Finally, by designing a similarity matrix for the transportation network environment, we obtain the recognition results for the current environment and state. Experimental results demonstrate that our method outperforms competitive approaches with a mean average precision (mAP) value of 0.776, serving as a theoretical foundation for visual scene perception in transportation networks and enhancing their level of intelligence.

# INTRODUCTION

Intelligent and self-organizing transportation networks are a hot topic in the modern urban transportation system (*Zhang & Lu, 2020*). Through the use of advanced technology and data analysis, optimizing transportation mobility, improving transportation efficiency, reducing congestion and environmental pollution, and improving the sustainability of the transportation system will help build an intelligent and efficient urban system and provide more convenient and environmentally friendly ways for urban residents to travel.

Smart and self-organizing transportation networks utilize technologies such as sensors, cameras and on-board equipment to collect transportation data and monitor transportation flow, congestion, transportation accidents and more in real time in order to better manage transportation mobility. For example, the cycle of transportation lights can be adjusted to optimize transportation flow and reduce congestion (*Yan, Liu & Tseng, 2020*; *Goswami et al., 2021*), provide drivers with real-time transportation information, suggest the best routes, reduce travel time and fuel consumption, and improve transportation safety and mobility through perception, decision making and control

Corresponding author
Jing Xu, xujing810904@outlook.com

systems that enable vehicles to travel without the need for human intervention (*Li et al., 2020*). In addition, in order to provide better transportation services, residents will have a lot of self-organizing behaviors. Shared cars, shared bikes and electric scooters, *etc.*, are distributed and scheduled through applications. Vehicles share information, such as transportation flow, road conditions, *etc.*, through communication networks in order to better plan driving paths and avoid congestion (*Liu et al., 2023*). Based on user needs, passengers on the same route can be automatically matched to realize shared carpooling and reduce the number of vehicles and transportation pressure. Through sensors and apps, it can guide drivers to find available parking spaces and reduce the time spent searching for parking spaces (*Schmid, Schraudner & Harth, 2021*).

These capabilities rely on the Internet of Things (IoT), big data analytics, and artificial intelligence technologies. By connecting various sensors and devices, real-time visual scene data is collected to provide data support for intelligent transportation systems. Artificial intelligence algorithms are used to predict the traffic flow, optimize the cycle of signal lights, improve the navigation system, *etc.*, and improve the intelligence level of the system (*Kaffash, Nguyen & Zhu, 2021*; *Arthurs et al., 2021*; *Haydari & Yilmaz, 2020*; *Lv, Lou & Singh, 2020*). *Vickrey (1969)* proposed a spatial pyramid network to divide images into regions and calculate local feature histograms within these sub-regions, thereby improving the ability to capture local image information. *Cheng, Pang & Pavlou (2020)* proposed an image representation based on semantic segmentation, which can not only classify images, but also specify the position of objects in a given scene according to their shape and size. *Wang et al. (2022)* employ the idea of transfer learning, using the previously trained model to transfer the learned parameter values to the new model. *Kang, Hu & Levin (2022)* used the CNN network and paid attention to the feature map output of the intermediate convolutional layer to generate classification features, and used a feature matching algorithm to improve the classification speed.

However, existing methods are all tailored to specific applications in the traffic environment, such as route selection, traffic bottlenecks, and traffic flow, without yet evolving into a unified multimodal traffic processing approach. Faced with increasingly complex traffic environments, unpredictable and uncertain scenarios have become more frequent. We propose a visual scene feature clustering analysis method based on 3D sensors and adaptive fuzzy control to address complex states in traffic networks and enhance their level of intelligence. This method aims to achieve precise identification and efficient processing of various complex scenarios in traffic networks through the integrated use of high-precision 3D sensor data and efficient adaptive fuzzy control algorithms. The 3D sensors capture detailed three-dimensional information in the traffic environment, including vehicle positions, speeds, directions, and pedestrian behavior patterns, providing abundant data sources for subsequent feature extraction and clustering analysis. On this basis, an adaptive fuzzy control algorithm is introduced to cope with the high degree of uncertainty and dynamic changes in the traffic environment. This algorithm can dynamically adjust the rules and parameters in the fuzzy logic system based on real-time traffic data, thereby enabling real-time, intelligent monitoring and response to traffic conditions. Through clustering analysis of visual scene features, we can identify bottleneck

areas, high-traffic road segments, and potential safety hazards within the traffic network, providing a scientific basis for traffic management and planning. Main contributions of our article are as follows:

We propose an improved method for visual scene feature extraction, which suppresses shadows and wrinkles through low-texture information processing and anti-aliasing contour modeling, to achieve the capture of 3D sensor information.

Also we propose a feature clustering method based on an adaptive fuzzy control algorithm, which utilizes extracted features for categorical detection and compensates for errors, realizing intelligent control of the traffic network.

# RELATED WORKS

## Research on visual scene features

Visual scene feature is a research topic used to clarify the data representation in a specific scene, which runs through the scene analysis task. The study of visual scene expression can be developed from many levels.

From the perspective of feature, scene feature is one of the most effective ways to simplify the expression of high-dimensional image data. In the fields of machine learning and pattern recognition, it is common for scholars to use features to describe measurable properties in the data to be processed. Due to the different forms of scenes and objects in scenes at the three levels of instance, category and semantics, the study of feature representation is usually faced with great difficulties.

*Zhang et al. (2021)* proposed a new weakly supervised representation learning method based on audio-visual scene analysis to construct a new multi-modal framework to instantiate learning, which has great application prospects in spatio-temporal visual positioning and other aspects. *Musaddiq, Zikria & Kim (2020)* designed a multi-modal fusion structure for the scene recognition task. The system can simultaneously identify the inter-modal and intra-modal information associations, and reverse propagate the results to the lower CNN layer, so as to effectively update the parameters of the CNN layer and the multi-modal layer. *Lu et al. (2020)* proposed a deep learning framework based on audio-visual data, focusing on the study of five types of riots and crowded scenes, such as movement atmosphere, noisy streets, riots, concerts, fireworks parties, *etc.*, and achieved an accuracy rate of 95.7%. *Nagatani & Hino (2015)* proposed a conditional attention fusion strategy for continuous dimension emotion prediction. The model can dynamically focus on different modal information according to current modal characteristics and historical information, thus improving the stability of the model. *Kessler & Bogenberger (2019)* breaks through the previous method of capturing context information through multi-scale feature fusion, and proposes a dual attention network that adaptively combines local features with their global dependence, and captures rich context information through self-attention mechanism to solve scene segmentation. *Wang, Wu & Wang (2020)* proposed an attention-guided point cloud and multi-view fusion network for 3D shape recognition, which enhanced the features of each mode by fusing the relevant information between the two modes through the attention model. *Wu et al. (2021)* proposes a general strategy for multi-model and multi-label classification tasks based on Transformer, which uses the

common attention mechanism to obtain the similarities between features to select better feature combinations (*Basu et al., 2022*).

## Fuzzy control algorithm

The fuzzy control algorithm can not only highly simulate the implement and decision process, but also in the case of no need to establish accurate control object model (*Wahba, 2004*). Among them, adaptive fuzzy control can boost the performance of the controller, modify and perfect the fuzzy control rules by itself, and has the ability of self-learning and self-adaptation. Especially for some complex systems such as nonlinear, high-order and large-time delay systems, the control effect of this algorithm is better (*Nguyen et al., 2019*). The fuzzy control combined with a variety of intelligent optimization algorithms can optimize the fuzzy control rules or membership function to achieve better control effects, and can constantly modify the control rules offline or online to improve the control accuracy of the system. Multivariable fuzzy control means that it contains multiple input and output variables, and this method has a good effect when applied to multivariable control systems (*Liang et al., 2020*). In addition, the fuzzy control rules and membership function of neural network can be modified and optimized by relying on its unique learning ability (*Precup et al., 2020*).

*Miranda et al. (2022)* proposed a kind of fuzzy logic decision, which can improve the effective dynamic performance of the vehicle in actual driving by distributing the motor power. This kind of fuzzy logic is the layered development of the controller to ensure that the speed of the car varies with the working conditions, which is equivalent to a differential to ensure that the car is always driving in the correct path and track. *Moudoud, Aissaoui & Diany (2022)* proposed an adaptive fuzzy synovial controller, which can track the trajectory of mobile robots with uncertainty and disturbance. *Mitra, Dey & Mudi (2021)* took a soccer cart as the controlled object of the experiment, applied fuzzy PID control technology to analyze its kinematics model, and proposed a fuzzy PID controller design method based on error partition in view of practical problems in motion control. The method not only increases the adjustability of the controller but also reduces the workload of the controller design. *Ahmed (2021)* used the linear quadratic controller to control the automobile vibration, and compared the performance with that of ordinary suspension, they concluded that the fuzzy PID control effect was better. *Khodadadi, Soleimanian Gharehchopogh & Mirjalili (2022)* proposed a multi-objective optimization algorithm based on the lifestyle of the African vulture, achieving good results in the mechanisms of archive management, grid partitioning, and leader selection. *Sharma et al. (2023)* introduced a butterfly optimization algorithm incorporating dominance sorting and crowding distance mechanisms to address various problems characterized by linear, nonlinear, continuous, and discrete features. *Ayar et al. (2023)* presented a multi-objective, non-dominated sorting competitive algorithm (NSICA) to tackle the optimal feature selection problem, which was applied in a feature selection system for diagnosing arrhythmias.

# VISUAL SCENE FEATURE AGGREGATION ANALYSIS BASED ON 3D SENSOR AND ADAPTIVE FUZZY CONTROL ALGORITHM

The visual scene feature clustering analysis method based on 3D sensors and adaptive fuzzy control primarily consists of the following core components: visual scene feature extraction using 3D sensors, feature clustering based on adaptive fuzzy control algorithms, and a traffic environment similarity calculation method. During the visual scene feature extraction stage using 3D sensors, we utilize high-precision 3D sensor equipment, such as Light Detection and Ranging (LiDAR) or depth cameras, to capture three-dimensional spatial information in the traffic environment. This information encompasses the precise positions, speeds, directions, sizes of vehicles, as well as the dynamic changes of pedestrians and obstacles, providing a detailed data foundation for subsequent feature analysis and clustering. Subsequently, we employ feature clustering techniques based on adaptive fuzzy control algorithms to intelligently classify the extracted visual scene features. This step effectively addresses the complex and variable challenges in the traffic environment by adaptively adjusting the rules and parameters in the fuzzy logic system, achieving precise identification and efficient classification of traffic conditions. The clustering results not only reveal key patterns in traffic flow but also provide powerful support for subsequent traffic management and decision-making. Lastly, we introduce a traffic environment similarity calculation method to assess the degree of similarity between different traffic scenarios. This method compares traffic characteristics across different periods or locations, combining historical data and real-time monitoring information, to provide traffic managers with important insights for predicting future traffic conditions, optimizing traffic signal control, and formulating emergency response strategies.

## Feature extraction of visual scene based on 3D sensor

### Data preprocessing

3D visual scene feature is the fundamental in the computer vision, aiming to capture information about shape, texture, structure and other aspects from 3D scene data to support applications such as scene analysis, object recognition and target tracking.

A typical 3D sensor (D435) is shown in Fig. 1, whose composition from left to right is left infrared camera, infrared dot matrix projector, right infrared camera and red green blue (RGB) camera. Among them, the RGB camera is used to directly acquire color images, while the left and right infrared cameras can receive infrared images and realize distance measurement by using the binocular ranging principle to obtain depth images. The infrared dot matrix projector in the middle is equivalent to a fill light, which is used to provide infrared patterns in the environment of low texture to improve the accuracy of binocular matching. For binocular ranging, the expression of distance from the measured object to the camera baseline is:

$$z = \frac{fB}{X_R - X_r} \tag{1}$$

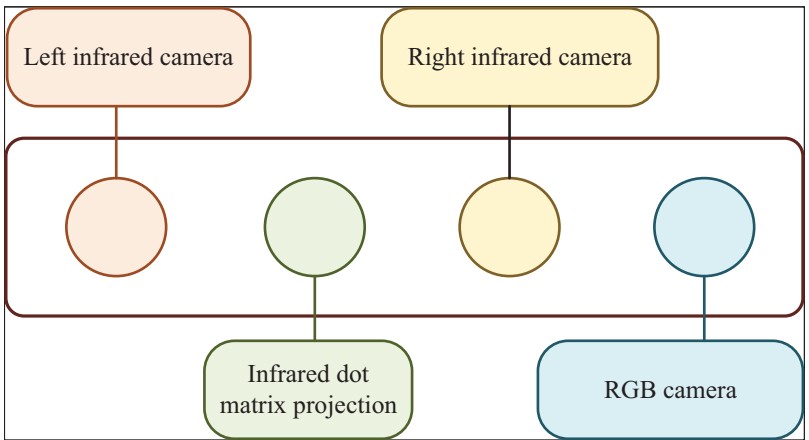

**Figure 1 Structure of 3D sensor.**

where focal length f and center distance B are both fixed camera parameters. Therefore, by obtaining parallax, which is XR-Xr, the distance can be calculated.

For any visual information I collected by the 3D sensor, this article performs a series of pre-processing on it. First of all, this article uses the close operation to remove the black noise of the image, so that the outline of the image becomes smooth, which can bridge the narrow discontinuity and slender gully in the image, eliminate the hole and fill the crack in the outline. Use element C to close the set AI, the formula is as follows:

$$A_I \cdot C = (A_I \oplus C) \ominus C. \tag{2}$$

Then, this article uses corrosion, Gaussian blur, and dilation operators to eliminate the jagged outline present at the edge of the target in the image. Finally, this article use the Canny to extract the object edge in the 3D scene and expand it to ensure the edge continuity. The core formula of the Canny operator is as follows:

$$H_{i,j} = \frac{1}{2\pi\sigma^2} exp\left(-\frac{(i-(k+1))^2 + (j-(k+1))^2}{2\sigma^2}\right) \tag{3}$$

where $1 \leq i, j \leq 2k+1$

Considering the data collected by sensors contains noise such as fine lines and isolated points, the opening-closing algorithm, which is based on morphological operations, has a relatively simple calculation process and is easy to implement. It demonstrates significant effectiveness in removing specific types of noise. Convolutional neural networks (CNNs) and generative adversarial networks (GANs) typically require a large amount of training data and computational resources, and their real-time performance is difficult to surpass that of the opening-closing algorithm. Additionally, we have adopted an adaptive algorithm in subsequent steps that can be integrated with the opening-closing algorithm.

### Feature extraction

After obtaining the pre-processed 3D image, this article takes the outermost outline as the principle to obtain the outline of the target in the image, to ignore the small internal

outline. The purpose of this is to prevent the shadow and wrinkles in the target from being detected, which will interfere with the subsequent image processing. At the same time, the smallest external positive rectangle corresponding to the contour is found to lock the area of interest (ROI) for subsequent calculation. The calculation of the ROI centroid is based on the contour and obtained by calculating the image moment. Only geometrical moments of order 0 and order 1 are needed to calculate the centroid. For an M × N image, the formula for the geometrical moments is:

$$m_{i,j} = \sum_{x=1}^{N} \sum_{y=1}^{M} f(x,y) \times x^i \times y^j \tag{4}$$

where i, j represent the order of geometric moments, and x, y represent pixels in the corresponding image. f(x, y) is the equivalent mass size corresponding to the point (x, y) in the image. Assuming f(x, y) is identical to a constant, then the centroid of the profile is the centroid of the profile, and the centroid coordinates are:

$$(x,y) = \left( \frac{m_{10}}{m_{00}}, \frac{m_{01}}{m_{11}} \right). \tag{5}$$

Thus, as shown in Fig. 2, the features of the objects in the visual scene of the transportation network can be obtained.

## Feature clustering based on adaptive fuzzy control algorithm

A fuzzy control algorithm that can carry out its structure in the design process needs to effectively combine the relationship between system structure and control algorithm. The fuzzy control contains various modules, among which the fuzzy interface module makes clear the fuzzy value by fuzzy processing the control input. Finally, the defuzzification converts the fuzzy output to the precise value. This article studies the fluency recognition algorithm of transportation network in complex environments. Through many 3D vision modules, the corresponding environment is classified and detected, and then the visual information is fused and effectively classified. By the error compensation method, the specific fuzzy control input can be formulated, and then this article can complete the reasoning and make a judgment. Finally, through the function of the defuzzification, the real transportation can achieve intelligent functions.

This article uses fuzzy means clustering (FMC), a soft clustering algorithm, to identify the information in the 3D sensor. In the process of clustering the samples, FMC algorithm does not feed the sample to a certain category. Through the fuzzy theory, the sample is divided into multiple classes and then divides each sample in the sample set into different clusters according to the membership degree. By constantly updating each cluster center, the function reaches the minimum value. Suppose the sample set is X = {x1, x2,…, xn}, and divide samples into class p with cluster center C. $\mu_{ik}$ indicates that sample xk belongs to the membership value of cluster center $c_i$, and the size of the membership value meets the following conditions:

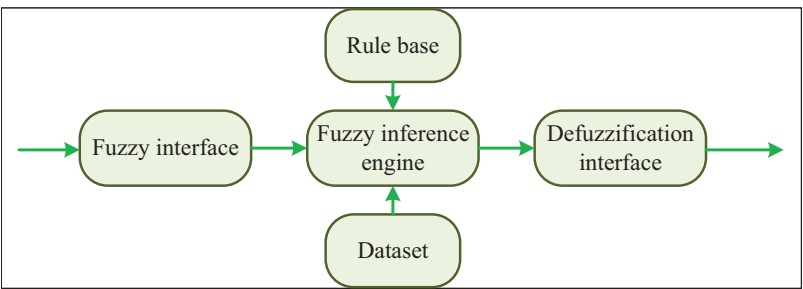

**Figure 2** **The structure of fuzzy control algorithm.**

$$\sum_{i=1}^{p} \mu_{ik} = 1, \; \forall k = 1, \ldots, n. \tag{6}$$

After clustering by FMC algorithm, p cluster center vectors and an n×p dimension fuzzy membership matrix U will be output.

The FMC algorithm has two important parameters. One is the number of cluster categories p, and the other is the fuzzy index m, which is used to adjust the fuzzy degree of the clustering model, usually the value of m is 2, whose functions are presented in the equations:

$$J(U, C) = \sum_{k=1}^{n} \sum_{i=1}^{p} (\mu_{ik})^m (d_{ik})^2 \tag{7}$$

$$(d_{ik})^2 = ||x_k - c_i|| \tag{8}$$

J(U, C) describes the distance from all samples to all cluster centers, where U denotes the membership and C refers to the center matrix. In addition, the updated formula for the membership matrix and cluster center is:

$$\mu_{ik} = \frac{1}{\sum_{j=1}^{n} \left(\frac{d_{ik}}{d_{jk}}\right)^{\frac{2}{m-1}}} \tag{9}$$

$$c_i^{(i+1)} = \frac{\sum_{k=1}^{n} \mu_{ik}^{(i+1)} x_k}{\sum_{k=1}^{n} \mu_{ik}^m}, \; i = 1, 2, \ldots, p. \tag{10}$$

The algorithm dynamically adjusts the input and output variables of the fuzzy logic system, as well as the fuzzy relationships between them, by collecting and analyzing real-time traffic data (such as vehicle flow, speed, direction, *etc.*). As the traffic environment changes, the algorithm can automatically optimize the fuzzy rule set to better reflect the current traffic conditions. Additionally, the algorithm can adjust the clustering strategy in real-time to adapt to new traffic characteristics. This includes adjusting the number and position of cluster centers, as well as the parameters of the clustering algorithm.

## Compute similarity for transportation environment

Similarity calculation is the core of perception of visual environment, which directly affects the judgment of transportation environment and the efficiency of algorithm. To calculate the similarity, this article conducts weighted summation by calculating the similarity of different transportation environments, which not only includes the explicit information of the visual scene in the sensor, but also considers the implicit information of the influence of people on transportation.

In this article, Pearson's similarity calculation method based on complex environment is used to obtain the similarity of the current environment through the clustering results of different transportation, as shown in the formula:

$$sim(u, v) = \frac{\sum\limits_{i \in u \cap v} \left(h_u - \overline{h_u}\right)\left(h_v - \overline{h_v}\right)}{\sqrt{\sum\limits_{i \in u \cap v} \left(h_u - \overline{h_u}\right)^2} \sqrt{\sum\limits_{i \in u \cap v} \left(h_v - \overline{h_v}\right)^2}}. \tag{11}$$

Considering the influence of people on transportation, this article introduces a penalty factor, as shown in the formula:

$$Q = \frac{1}{1 + \log\left(1 + \frac{|N_c|}{n}\right)} \tag{12}$$

where n represents the number of different environments. Then Formula (11) can be amended to:

$$sim(u, v) = \frac{\sum\limits_{i \in u \cap v} \left(h_u - \overline{h_u}\right)\left(h_v - \overline{h_v}\right) \cdot Q}{\sqrt{\sum\limits_{i \in u \cap v} \left(h_u - \overline{h_u}\right)^2} \sqrt{\sum\limits_{i \in u \cap v} \left(h_v - \overline{h_v}\right)^2}}. \tag{13}$$

The complex environment similarity matrix can be obtained through similarity calculation. The greater the similarity value between environments, the closer they are. In this article, the complex environment is divided into different clusters by the improved fuzzy clustering method. In the same cluster, the sample with the highest similarity value to the target environment is selected as the nearest neighbor set, denoted as N, then the probability distribution of the current transportation environment u can be expressed as:

$$P = \overline{r_u} + \frac{\sum\limits_{v \in N} sim(u, v)(r_v - \overline{r_v})}{\sum\limits_{v \in N} sim(u, v)}. \tag{14}$$

## EXPERIMENT AND ANALYSIS

### Dataset and implement details

This article uses the Transportation dataset (DOI 10.5281/zenodo.1205229) to test based on 3D sensor and adaptive fuzzy control algorithm of the visual scene clustering method.

This dataset contains urban traffic speed information from 214 anonymous road segments (primarily comprising urban expressways and arterials) over a 2-month period, spanning 61 days from August 1, 2016, to September 30, 2016, with data recorded at 10-min intervals. The speed observations were gathered in Guangzhou, China. It is practically applicable for experiments involving missing data imputation, short-term traffic prediction, and the discovery of traffic patterns.

The model training parameters for the entire experiment are shown in Table 1. The experiment was conducted on an Ubuntu system, using Python as the programming language. Additionally, the learning rate for model training was set to 0.001, with a decay rate of 0.92 and a momentum of 0.95. The mean absolute error (MAE) was adopted as the training optimizer.

Taking into account consumer preferences, therefore use the mean average precision (mAP) to evaluate the method, which is calculated by the equations:

$$Precision = \frac{V(gt \bigcap pr)}{V(pr)} \tag{15}$$

$$Recall = \frac{V(gt \bigcap pr)}{V(gt)} \tag{16}$$

$$mAP = \frac{1}{N} \times \sum Precision_n \times Recall_n \tag{17}$$

where pr refers to the predicted result and gt denotes the truth of the data set annotation.

## Comparison of our method and other methods

This article demonstrated the performance experiment of our method on the Transportation dataset. This article selected some excellent feature models, such as Binary tree (Huang et al., 2019), CNN (Kattenborn et al., 2021), Transformer (Bagal et al., 2021), Bert (Deepa, 2021), Vit (Khan et al., 2022), TypeFormer (Stragapede et al., 2022), EfficientNet (Tan & Le, 2019), You Only Look Once (YOLO)v5 (Kim et al., 2022) and Swin Transformer (Liu et al., 2021) and compared the performance, as shown in Fig. 3. This article can find that compared with other transportation environment recognition algorithms, our method can obtain the highest mAP value, which is 0.776. Compared with the binary tree, our method improves mAP value by more than 9%. The reason for this improvement is that the binary tree is better at handling binary classification problems. The complex transportation environment cannot be simply divided by this linear way. Compared to CNN, EfficientNet and YOLOv5, our method still has a lead of more than 5%. Unlike the binary tree, which does not pass, CNN has the ability to handle multiple categories, but it does not have good fusion performance when dealing with multi-feature tasks. Self-attention-based Transformer and Bert are capable of handling multi-modal tasks in complex environments, but compared with our approach, it improves by more than 4%. This is because our approach is based on the concept of fuzzy control, which enables dynamic recognition of complex environments, rather than obtaining a single probability distribution. Swin Transformer, Vit and Typeformer are the latest proposed classification methods. They can obtain mAP values above 0.73 with excellent model

**Table 1 Implementation details.**

| Types | Parameters |
|---|---|
| Computer detail | R7-7900xt |
| GPU | Rtx 2080Ti |
| Framework | Caffe |
| Training epochs | 45 |
| Batch size | 32 |
| Initial learning rate | 0.001 |

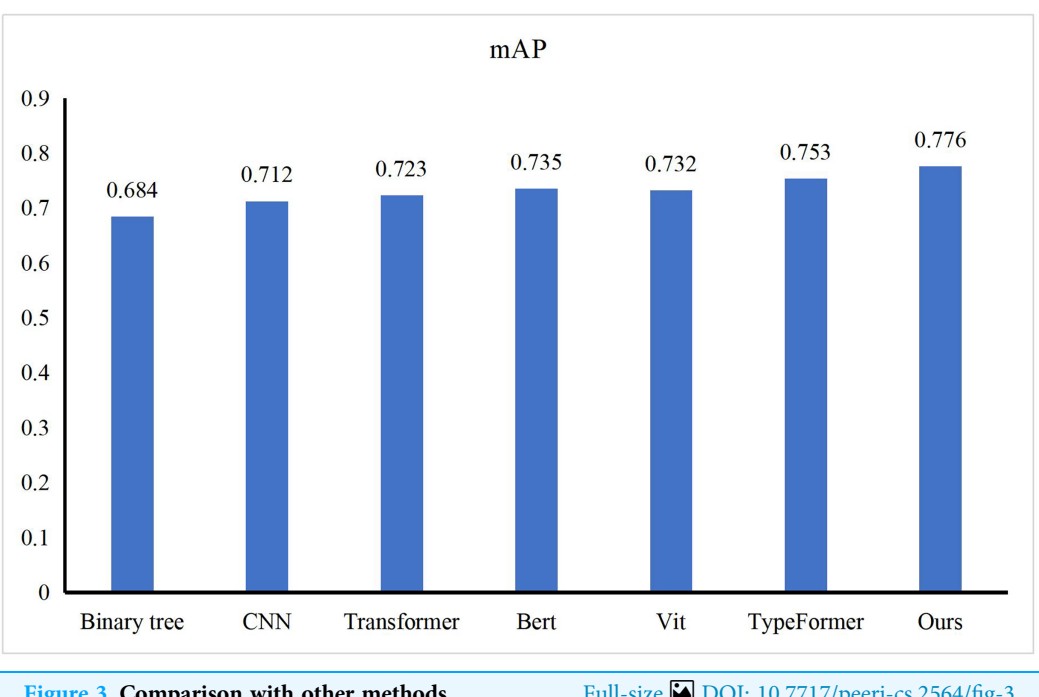

**Figure 3 Comparison with other methods.**

performance, and our method is still more than 2% higher than them. By analyzing the complex scenes that may appear in the transportation network, our method designs the scene recognition method and gives up the fixed category setting in the classification method to obtain more efficient recognition results.

Furthermore, leveraging the structural characteristics of fuzzy means clustering, we conducted a comparative analysis of our model's training process with other models such as Bert, Vit, and Typeformer. As illustrated in Fig. 4, the model's loss function curve clearly demonstrates that our model has essentially reached a state of fitting by the 35th training epoch, indicating that it has obtained the optimal parameter configuration at this point. This remarkable convergence speed is attributed to the efficient optimization algorithms and advanced model structure we employed, which collectively facilitate the rapid search for global or near-global optimal solutions during the training process. Additionally, we implemented effective regularization techniques and early stopping strategies to prevent

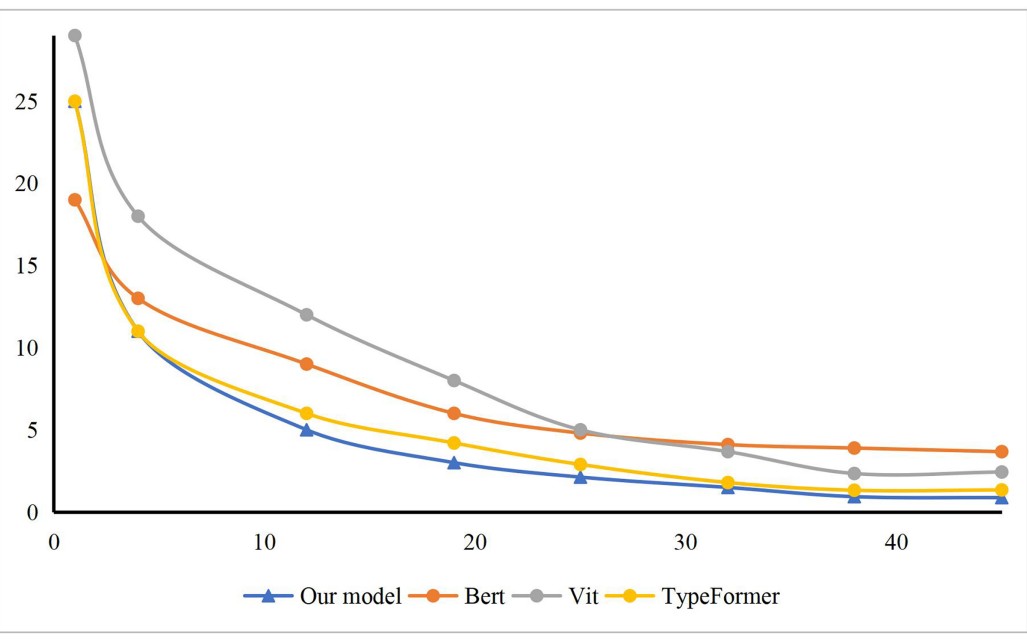

**Figure 4** **The loss of our model comparing with others.**

overfitting, which not only enhanced the model's generalization ability but also further accelerated the convergence process. Compared to other models, our model not only excels in convergence speed but also boasts a more streamlined and efficient structure. By simplifying model complexity and eliminating redundancy, we significantly reduced computational costs and improved training efficiency. Consequently, our model converges to the optimal state faster during the initial stages of training, demonstrating its superior performance and efficient training characteristics. To demonstrate the superiority of our method in terms of time loss and memory consumption, we conducted comparisons with other methods. The experimental results, presented in Fig. 5, show that our method is leading in terms of time costing, model parameter, and GPU memory usage. The entire process took 12 ms, occupied 2.2 GB of GPU memory, and had a model parameter count of 14.22 million. The method closest to ours is the binary tree approach, but it exhibits significantly poorer performance compared to ours. The other methods consumed considerably more computational resources.

## The influence of visual scene feature processing

During the feature extraction phase, we applied closing operation, erosion, Gaussian blur, and dilation processing techniques to the features of 3D visual scenes. These image processing operators effectively repaired cracks and noise in 3D images. To investigate the impact of cracks and noise on the recognition of complex traffic network environments, we designed a series of ablation experiments specifically to assess the effects of closing operation (CO), erosion (C), Gaussian blur (GB), and dilation (E). Throughout the ablation experiments, we maintained consistency in the feature extraction process and subsequent processing.

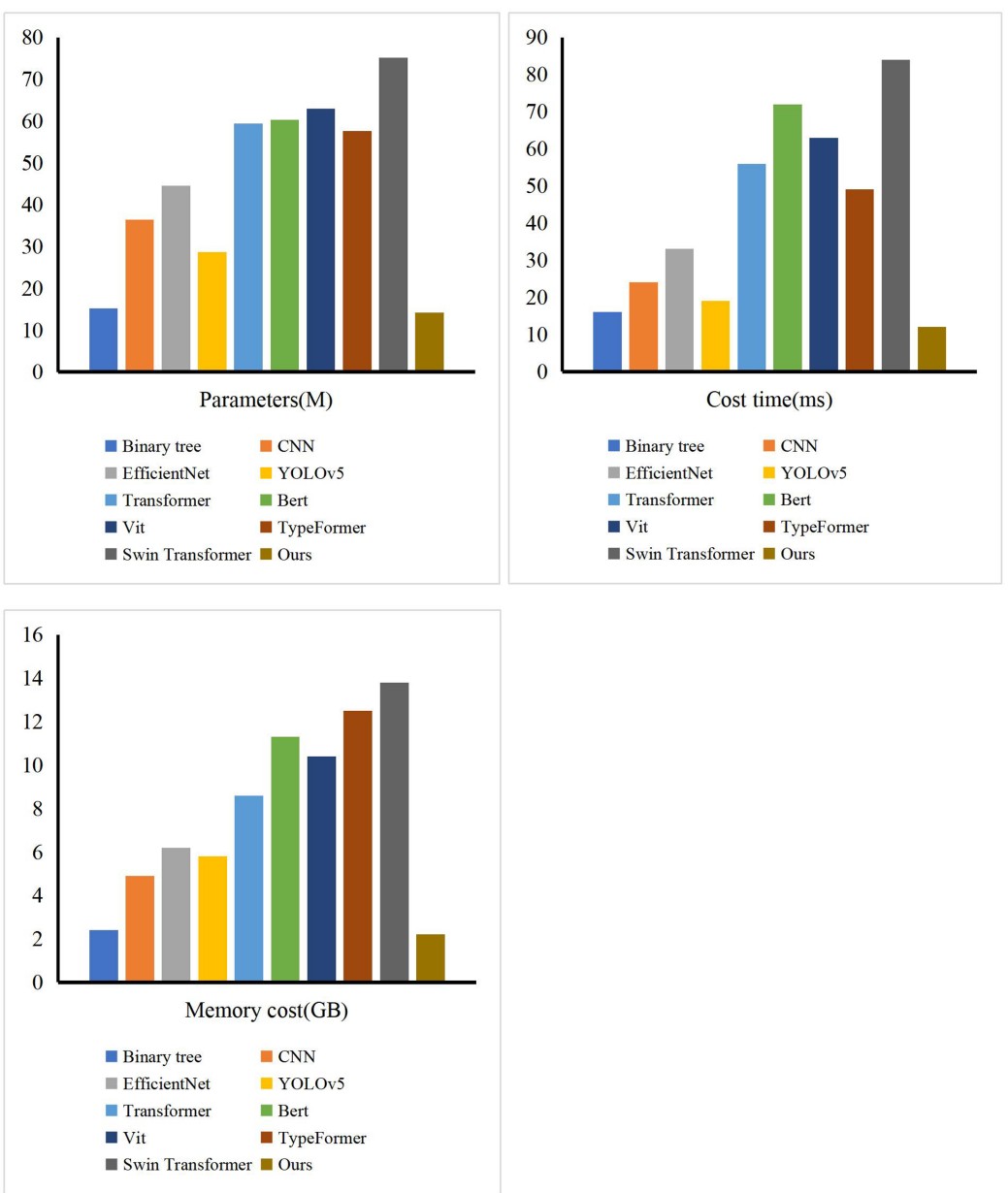

**Figure 5 The comparison of our model and others in terms of time cost, model parameters and memory cost.**

The experimental results, as shown in Table 2, indicate that sequentially applying closing operation, erosion, Gaussian blur, and dilation processing to 3D images can gradually and significantly improve the recognition accuracy of complex traffic network environments. When CO, C, GB, or E was individually introduced, the model's mAP reached at least 0.71. As other processing modules were gradually integrated into the model, the performance of the model improved by at least 1% each time. Ultimately, the model we constructed achieved a high performance level of 0.776. This is because these processing methods can make up for the error of the data collected by the 3D sensor and

**Table 2 Ablation of CO, C, GB and E.**

| CO | C | GB | E | mAP |
|---|---|---|---|---|
| ○ | | | | 0.718 |
| | ○ | | | 0.721 |
| | | ○ | | 0.714 |
| | | | ○ | 0.720 |
| ○ | ○ | | | 0.733 |
| ○ | | ○ | | 0.739 |
| ○ | | | ○ | 0.735 |
| | ○ | ○ | | 0.740 |
| | ○ | | ○ | 0.735 |
| | | ○ | ○ | 0.743 |
| ○ | ○ | ○ | | 0.756 |
| ○ | | ○ | ○ | 0.759 |
| | ○ | ○ | ○ | 0.763 |
| ○root | ○ | ○ | ○ | 0.776 |

highlight the main content of the environmental image. In addition, it also shows that the inevitable cracks and noise of 3D images are factors that affect the environment recognition in the transportation network, which will directly hinder the intelligent process of the transportation network.

## Discussion

This article proposes a visual scene feature cluster analysis method based on 3D sensors and adaptive fuzzy control to realize complex environment recognition of transportation networks and improve their intelligence. The above experiments can conclude that our model has excellent mAP value and can be the theoretical basis of the intelligent transportation network. Among them, the 3D sensor can capture the three-dimensional spatial information of the object, rather than just the two-dimensional image. They can provide richer and more accurate scene information, helping to better understand objects and structures in the environment. Considering the complex scenario of the transportation network, the adaptive fuzzy control this article adopted, a logic used to deal with uncertainty and fuzziness, allows the control system to adjust according to the real-time situation, which can be used to optimize transportation signal control, vehicle path planning, *etc*. Finally, the location, size, shape and other features of objects can be extracted from the 3D scene through the aggregation analysis of visual scene features, and even target recognition and tracking may be carried out.

Through the research of the above methods, 3D sensor-based visual scene feature analysis can help vehicles and infrastructure to better perceive the surrounding environment. For example, vehicles can identify obstacles in front of them, pedestrians, *etc*., to make appropriate driving decisions. With the help of adaptive fuzzy control algorithms, transportation signal control can be adjusted according to real-time

transportation flow and scene information, thereby reducing congestion and optimizing transportation flow. In addition, 3D sensors can help identify potentially dangerous situations, such as fast-approaching vehicles or sudden obstructions. Adaptive fuzzy control can prompt a vehicle to take actions to avoid a collision. With environmental information obtained from sensors, vehicles can choose smarter paths and avoid congested or dangerous areas, thus improving efficiency and safety.

To sum up, the clustering analysis of visual scene features based on 3D sensors and adaptive fuzzy control algorithms can play an important role in intelligent transportation networks. The integration of these technologies can help create a more intelligent and highly adaptive transportation system.

## CONCLUSION

To enhance the intelligence of transportation networks, this article introduces a visual scene feature clustering analysis method based on 3D sensors and adaptive fuzzy control. By extracting visual features from 3D sensor-based scenes and applying preprocessing algorithms, noise within the 3D scenes is removed, and primary targets are highlighted. Leveraging the visual features of transportation network scenes, an adaptive fuzzy control-based feature aggregation model is designed to achieve in-depth processing of 3D scene features. By constructing similarity matrices for various environments, we obtain recognition results for transportation network environments. Experimental results demonstrate that our method achieves an mAP value of 0.776, surpassing current state-of-the-art algorithms. Furthermore, ablation experiments highlight the necessity of suppressing cracks and noise in 3D transportation network scenes.

## ACKNOWLEDGEMENTS

The author would like to thank the anonymous reviewers who have provided valuable comments on this article.

### Funding
The author received no funding for this work.

### Competing Interests
The author declares that they have no competing interests.

### Author Contributions
- Jing Xu conceived and designed the experiments, performed the experiments, analyzed the data, performed the computation work, prepared figures and/or tables, authored or reviewed drafts of the article, and approved the final draft.

### Data Availability
The code is available in the Supplemental File.

The third-party dataset is available at Zenodo: Xinyu Chen, Yixian Chen, & Zhaocheng He. (2018). Urban Traffic Speed Dataset of Guangzhou, China [Data set]. Zenodo. https://doi.org/10.5281/zenodo.1205229.

## Supplemental Information

Supplemental information for this article can be found online at http://dx.doi.org/10.7717/peerj-cs.2564#supplemental-information.

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
