# Peer review of "Enhancing transportation network intelligence through visual scene feature clustering analysis with 3D sensors and adaptive fuzzy control"

_PeerJ Computer Science, doi:10.7717/peerj-cs.2564_

## Round 0.1 · original submission · Major Revisions

Based on the reviewers’ comments, you may resubmit the revised manuscript for further consideration. Please consider the reviewers’ comments carefully and submit a list of responses to the comments along with the revised manuscript.

Reviewer 1 ·

Basic reporting

It is hoped that the following suggestions will help the authors to improve the scientific and practical nature of the manuscript and ensure the validity and feasibility of the proposed method:

˙ In the introduction, the research background is more clearly stated, including the specific problems and challenges faced by the current transportation network and the impact of these problems on the transportation system. It should be specified why existing methods are insufficient to address these issues, and the motivation and purpose of the research should be clearly presented.

˙ The specific implementation details of the visual scene feature extraction and adaptive fuzzy control algorithm are vague in the current method description.

Experimental design

˙ The 3D sensor model, parameters, data acquisition methods, and specific image processing algorithms (such as denoising methods and feature extraction techniques) should be described in detail. The algorithm and parameter Settings of each step are specified in order to facilitate reproduction and verification.

˙ Comparing only with Binary tree, CNN, Transformer, Bert, Vit, and TypeFormer may have missed other state-of-the-art models. Add experiments against other modern architectures (e.g. EfficientNet, YOLOv5, Swin Transformer, etc.) to fully evaluate the relative performance of the methods.

˙ The working principle of the algorithm, the parameter adjustment method and its role in feature aggregation are described in detail, especially how to deal with dynamically changing environmental conditions.

Validity of the findings

˙ It is not mentioned that the diversity and size of the dataset used may affect the generalization ability of the method.

˙ Computational complexity and resource consumption of the model are not analyzed, which may affect practical applications. A detailed model complexity analysis, including computation time and memory consumption, is provided to evaluate the feasibility of the model in practical applications.

˙ The comparison of convergence rates is superficial and lacks a detailed analysis. A detailed convergence curve is provided, and the reasons for the faster convergence speed of the model are analyzed, including the learning rate setting, model structure and other factors.

Additional comments

˙ Make sure the language is formal and academic, and avoid colloquialisms or vague expressions. For example, changing "this method may be better" to "This method shows significant advantages in performance evaluation". Check spelling, grammar and word accuracy


˙ In the discussion section, the impact and significance of the results are systematically analyzed and compared with the literature in related fields. The strengths and weaknesses of the method are discussed, and suggestions for improvement or future research directions are present.

Reviewer 2 ·

Basic reporting

The feature aggregation model of adaptive fuzzy control is designed to realize the deep processing of 3D scene features by using the transportation network scene visual features. By constructing the similarity matrix of each environment, this paper can get the recognition result of the transportation network environment. However, the following suggestions are strongly recommended to be incoroporated.

1. The abstract section is fragile. Please rewrite it, explain the result obtained and contribution, improve a proposed method, and delete unnecessary information.
2. Proofread the manuscript carefully to eliminate any grammatical errors or typos and ensure clarity and coherence in writing. Additionally, adhere to the formatting and style guidelines specified by the target journal or publication venue to enhance the professionalism of the manuscript.
3. Please write your contribution to this paper in the Introduction section.
4. Divide the methods section into smaller subsections such as "Data preprocessing", "Feature extraction", "fuzzy control algorithm", "similarity matrix design", etc. Each section clearly describes its process and technical details
5. How did the authors set parameters for their proposed algorithm? Please make sensitivities of these parameters to the performance of their proposed algorithm!
6.Incorporating relevant and recent academic sources could strengthen your paper's validity and give readers more context and background. advances in manta ray foraging optimization: a comprehensive survey, moavoa: a new multi-objective artificial vultures optimization algorithm, non-dominated sorting advanced butterfly optimization algorithm for multi-objective problems, nsica: multi-objective imperialist competitive algorithm for feature selection in arrhythmia diagnosis,

Experimental design

7. Expand the critical results in the conclusion. Focus on the main developments in the finale. Also, write the main contributions in the conclusion.
8. Numerical results are good enough, but more explanations are required to analyze each figure presented.
9. Current image inpainting methods may not adequately address the problems of cracks and noise. Why not consider adopting advanced denoising techniques such as the denoising algorithms in Convolutional Neural Networks (CNNS) or Generative Adversarial Networks (GANs)?
10. The simulation section needs to be more detailed. The authors should provide more information about the data they employed and the simulation process.
11. Please improve the quality of some of the figures.

Validity of the findings

Plz check the above comments.

Additional comments

Plz check the above comments.

---

## Round 0.2 · accepted · Accept

Congratulations, the reviewers are satisfied with the revised version of the manuscript and recommended accept decision.

Reviewer 1 ·

Basic reporting

The authors have addressed all the concerns. The paper can be accepted in its current state.

Experimental design

NA

Validity of the findings

NA

Additional comments

NA

Reviewer 2 ·

Basic reporting

NA

Experimental design

No comment

Validity of the findings

no further comments

Additional comments

NA